# Are there identifiable structural parts in the sentence embedding whole?

**Vivi Nastase[1]  and  Paola Merlo[1,2]**
[1]Idiap Research Institute, Martigny, Switzerland
[2]University of Geneva, Swizerland
vivi.a.nastase@gmail.com, Paola.Merlo@unige.ch

## Abstract

Sentence embeddings from transformer models encode much linguistic information in a fixed-length vector. We investigate whether structural information – specifically, information about chunks and their structural and semantic properties – can be detected in these representations. We use a dataset consisting of sentences with known chunk structure, and two linguistic intelligence datasets, whose solution relies on detecting chunks and their grammatical number, and respectively, their semantic roles. Through an approach involving indirect supervision, and through analyses of the performance on the tasks and of the internal representations built during learning, we show that information about chunks and their properties can be obtained from sentence embeddings.

## 1 Introduction

Transformer architectures compress the information in a sentence – morphological, grammatical, semantic, pragmatic – into a fixed-length one-dimensional array of real numbers. Sentence embeddings, usually fine-tuned, have proven useful for a variety of high-level language processing tasks, such as the GLUE tasks (Clark et al., 2020), or story continuation (Ippolito et al., 2020)). These results, however, do not shed light on what kind of semantic or structural information is encoded in these representations.

Understanding what kind of information is encoded in the sentence embeddings, and how it is encoded, has multiple benefits. It connects internal changes in the model parameters and structure with changes in its outputs. It contributes to verifying the robustness of models and whether or not they rely on shallow or accidental regularities in the data. It narrows down the field of search when a language model produces wrong outputs, and ultimately it may help maximize the use of training data for developing more robust models from smaller textual resources. Investigation, or indeed, usage, of raw (i.e. not fine-tuned) sentence embeddings obtained from a transformer model are rare, possibly because most transformer models do not have a strong supervision signal on the sentence embedding. Using PCA analysis, Nikolaev and Padó (2023c) have shown that the dimensions of BERT sentence embeddings have much correlation and redundancy, and encode more shallow information (length), rather than morphological, syntactic or semantic features. Analysis of information propagation through the transformer layers seem to show that specialized information – e.g. POS, syntactic structure – while quite apparent at lower levels, gets lost towards the highest levels of the models (Rogers et al., 2020), while there are subnetworks that encode specific linguistic functions (Csordás et al., 2021; Conmy et al., 2023).

While previous work has regarded network nodes or embedding dimensions as the unit of analysis, Elhage et al. (2022) show that superposition – whereby each unit, i.e. neuron or embedding dimension, can be involved in the encoding of multiple features – occurs in artificial neural networks. Such features involving overlapping sets of nodes can be learned from a model using sparse autoencoders (e.g. (Cunningham et al., 2023)). Starting from a similar hypothesis relative to the dimensions of a sentence embedding, we aim to test whether specific information, in particular chunks – noun, verb and prepositional phrases, that may play different structural and semantic roles – can be detected in the sentence representation. We use an encoder-decoder architecture applied to data with specific properties, and verify that, through indirect supervision, we can distill information about chunks and their task-relevant properties from sentence embeddings from a pre-trained transformer model. Besides being practically useful, as they provide useful shallow structure more easily obtainable than detailed syntactic analysis (Abney, 1991; Buchholz

et al., 1999), chunks have psychological plausibility (Gee and Grosjean, 1983). This motivated us to test whether they are detectable in sentence embeddings, as they would provide syntactically and semantically useful building blocks for assembling higher level information about a sentence. The code and data are available at `https://github.com/CLCL-Geneva/BLM-SNFDisentangling`.

## 2  Related work

How is the information from a textual input encoded by transformers? There are three main approaches to answer this question: (i) tracing specific information from input to output through the model's various layers and components, (ii) isolating subsets of model parameters that encode specific linguistic functions and (iii) investigating the generated embeddings through probes, using purposefully built data for different types of testing.

**Tracing information through a transformer** Rogers et al. (2020) have shown that from the unstructured textual input, BERT (Devlin et al., 2019) is able to infer POS, structural, entity-related, syntactic and semantic information at successively higher layers of the architecture, mirroring the classical NLP pipeline (Tenney et al., 2019a). Further studies have shown that the information is not sharply separated, information from higher levels can influence information at lower levels, such as POS in multilingual models (de Vries et al., 2020), or subject-verb agreement (Jawahar et al., 2019). Surface syntactic and semantic information seem to be distributed throughout BERT's layers (Niu et al., 2022; Nikolaev and Padó, 2023c). Attention is part of the process, as it helps encode various types of linguistic information (Rogers et al., 2020; Clark et al., 2019), syntactic dependencies (Htut et al., 2019), grammatical structure (Luo, 2021), and can contribute towards semantic role labeling (Tan et al., 2018; Strubell et al., 2018).

**Isolating functional subnetworks of parameters** Deep learning models have billions of parameters. This makes them not only incomprehensible, but also expensive to train. The lottery ticket hypothesis (Frankle and Carbin, 2018) posits that large networks can be reduced to subnetworks that encode efficiently the functionality of the entire network. Detecting functional subnetworks can be done *a posteriori*, over a pre-learned network to investigate the functionality of detected subnetworks

(Csordás et al., 2021), the potential compositionality of the learned model (Lepori et al., 2023), or where task-specific skills are encoded in a fine-tuned model (Panigrahi et al., 2023). Instead of learning a sparse network over a prelearned model, Cao et al. (2021) use a pruning-based approach to finding subnetworks in a pretrained model that performs some linguistic task. Pruning can be done at several levels of granularity: weights, neurons, layers. Their analyses confirm previous investigations of the types of information encoded in different layers of a transformer (Conneau et al., 2018a). Conmy et al. (2023) introduce the Automatic Circuit DisCovery (ACDC) algorithm, which adapts subnetwork probing and head importance score for pruning to discover circuits that implement specific linguistic functions. The model network need not be separated into disjunct subsets of nodes. Elhage et al. (2022) show that neural network models encode more features than the number of their dimensions, individual nodes contributing to more than one feature. Such features could be learned in an unsupervised manner using Sparse AutoEncoders (Cunningham et al., 2023; Trenton Bricken, 2023; Gao et al., 2024), and correlated with linguistic patterns or phenomena.

**Word embeddings** were shown to encode sentence-level information (Tenney et al., 2019b), including syntactic structure (Hewitt and Manning, 2019), even in multilingual models (Chi et al., 2020). Predicate embeddings contain information about their semantic roles structure (Conia and Navigli, 2022), embeddings of nouns encode subjecthood and objecthood (Papadimitriou et al., 2021). The averaged token embeddings are more commonly used as *sentence embeddings* (e.g. (Nikolaev and Padó, 2023a)), or the special token ([CLS]/) embeddings are fine-tuned for specific tasks such as story continuation (Ippolito et al., 2020), sentence similarity (Reimers and Gurevych, 2019), alignment to semantic features (Opitz and Frank, 2022). Sentence embeddings as averages over token embeddings is justifiable as the learning signal for transformer models is stronger at the token level, with a much weaker objective at the sentence level – e.g. next sentence prediction (Devlin et al., 2018; Liu et al., 2019), sentence order prediction (Lan et al., 2019). Electra (Clark et al., 2020) relies on replaced token detection, which uses the sentence context to determine whether a (number of) token(s) in the given sentence were replaced by

a generator sample. This training regime leads to sentence embeddings that perform well on the General Language Understanding Evaluation (GLUE) benchmark (Wang et al., 2018) and Stanford Question Answering (SQuAD) dataset (Rajpurkar et al., 2016), or detecting verb classes (Yi et al., 2022). Raw sentence embeddings were shown to capture shallower information (Nikolaev and Padó, 2023c), but Nastase and Merlo (2023) show that raw sentence embeddings have internal structure that can encode grammatical sentence properties.

**Probing models**   Analysis of BERT's inner workings has been done using probing classifiers (Belinkov, 2022), or through clustering based on the representations at the different levels (Jawahar et al., 2019). Probing has also been used to investigate the representations obtained from a pre-trained transformer model (Conneau et al., 2018b). Elazar et al. (2021) propose amnesic probing to test both whether some information is encoded, and whether it is used. VAE-based methods (Kingma and Welling, 2013; Bowman et al., 2016) have been used to detect or separate specific information from input representations. Mercatali and Freitas (2021) capture discrete properties of sentences encoded with an LSTM (e.g. number and aspect of verbs) on the latent layer. Bao et al. (2019) and Chen et al. (2019) learn to disentangle syntactic and semantic information. Silva De Carvalho et al. (2023) learn to disentangle the semantic roles in natural language definitions from word embeddings. Probing can have issues: learning a classifier for a task does not guarantee that the model uses the targeted information (Hewitt and Liang, 2019; Belinkov, 2022; Lenci, 2023). Michael et al. (2020) introduce latent subclass learning, where a binary classification task has a pre-classification multi-class logistic regression step that helps probe for emergent information.

**Data**   Most approaches use datasets built by selecting, or constructing, sentences with specific structure and properties: definition sentences with annotated roles (Silva De Carvalho et al., 2023), sentences built according to a given template (Nikolaev and Padó, 2023b), sentences with specific structures for investigating different tasks, in particular SentEval (Conneau and Kiela, 2018) (Jawahar et al., 2019), example sentences from FrameNet (Conia and Navigli, 2022), a dataset with multi-level structure inspired by the Raven Progressive Matrices (RPM) visual intelligence tests (An et al., 2023).

# 3   Overview

Our approach is also a kind of probe. It uses indirect supervision, though, to avoid the shallow learning of a classifier and datasets with specific structure to test for structural information in sentence embeddings.

Our main object of investigation are chunks, sequence of adjacent words that segment a sentence, as defined initially in Abney (1992); Collins (1997) and then Tjong Kim Sang and Buchholz (2000). We use two types of data. We use **sentences with known chunk patterns** (Section 4.1), to determine whether chunks and their grammatical properties are identifiable in sentence embeddings with indirect supervision (Section 5). We also use **two datasets with multi-level structure** built for linguistic intelligence tests for language models (Merlo, 2023) (Section 4.2), to determine whether a system can detect syntactic and semantic structure and information in sentence embeddings based on the requirements of a task.

The data, with its repetitive patterns, and the VAE-based system support an indirect supervision approach: the system is not given the patterns to be discovered explicitly, but it needs to find them based on the contrasting answer sets at both the sentence and task levels. This indirect supervision process, together with lexical and structural variations in the data, helps to avoid, at least partly, the critiques against probes based on classification, which can learn a task based on 'artefacts' of the data, regularities different from what is intended (Belinkov, 2022).

# 4   Data

We use data consisting of stand-alone sentences with specific structure, and data consisting of sentences with specific structure and other attributes in larger contexts, to test whether this regular information can be detected.

## 4.1   Sentences

Sentences are built from a seed file containing noun, verb and prepositional phrases, including singular/plural variations. From these chunks, we built sentences with all (grammatically correct) combinations of np (pp$_1$ (pp$_2$)) vp[1]. For each chunk pattern $p$ of the 14 possibilities, all corresponding sentences are collected into a set $S_p$.

---

[1]We use BNF notation: pp$_1$ and pp$_2$ may be included or not, pp$_2$ may be included only if pp1 is included

**BLM agreement problem** (BLM-AgrF)

| CONTEXT TEMPLATE | | | |
|---|---|---|---|
| NP-sg | PP1-sg | | VP-sg |
| NP-pl | PP1-sg | | VP-pl |
| NP-sg | PP1-pl | | VP-sg |
| NP-pl | PP1-pl | | VP-pl |
| NP-sg | PP1-sg | PP2-sg | VP-sg |
| NP-pl | PP1-sg | PP2-sg | VP-pl |
| NP-sg | PP1-pl | PP2-sg | VP-sg |

| ANSWER SET | | | | |
|---|---|---|---|---|
| NP-pl | PP1-pl | PP2-sg | VP-pl | CORRECT |
| NP-pl | PP1-pl | et PP2-sg | VP-pl | Coord |
| NP-pl | PP1-pl | | VP-pl | WNA |
| NP-pl | PP1-sg | PP1-pl | VP-pl | WN1 |
| NP-pl | PP1-pl | PP2-pl | VP-pl | WN2 |
| NP-pl | PP1-pl | PP2-pl | VP-sg | AEV |
| NP-pl | PP1-pl | PP2-pl | VP-sg | AEN1 |
| NP-pl | PP1-pl | PP2-sg | VP-sg | AEN2 |

**BLM verb alternation problem** (BLM-s/lE)

| CONTEXT TEMPLATE | | | |
|---|---|---|---|
| NP-Agent | Verb | NP-Loc | PP-Theme |
| NP-Theme | VbPass | PP-Agent | |
| NP-Theme | VbPass | PP-Loc | PP-Agent |
| NP-Theme | VbPass | PP-Loc | |
| NP-Loc | VbPass | PP-Agent | |
| NP-Loc | VbPass | PP-Theme | PP-Agent |
| NP-Loc | VbPass | PP-Theme | |

| ANSWER SET | | | | |
|---|---|---|---|---|
| NP-Agent | Verb | NP-Theme | PP-Loc | CORRECT |
| NP-Agent | *VbPass | NP-Theme | PP-Loc | AGENTACT |
| NP-Agent | Verb | NP-Theme | *NP-Loc | ALT1 |
| NP-Agent | Verb | *PP-Theme | PP-Loc | ALT2 |
| NP-Agent | Verb | *[NP-Theme | PP-Loc] | NOEMB |
| NP-Agent | Verb | NP-Theme | *PP-Loc | LEXPREP |
| *NP-Theme | Verb | NP-Agent | PP-Loc | SSM1 |
| *NP-Loc | Verb | NP-Agent | PP-Theme | SSM2 |
| *NP-Theme | Verb | NP-Loc | PP-Agent | AASSM |

Figure 1: Structure of two BLM problems, in terms of chunks in sentences and sequence structure. For the **agreement** (left): (i) sequence errors: WNA= wrong nr. of attractors; WN1= wrong gram. nr. for $1^{st}$ attractor noun (N1); WN2= wrong gram. nr. for $2^{nd}$ attractor noun (N2); (ii) grammatical errors: AEV=agreement error on the verb; AEN1=agreement error on N1; AEN2=agreement error on N2. For the **verb alternation**: AGENTACT, ALT1, ALT2, NOEMB are syntactic errors; LEXPREP is lexical selection error and SSM1, SSM2, AASSM are syntax-semantic mapping errors.

We generate an instance for each sentence $s$ from the sets $S_p$ as a triple $(in, out^+, Out^-)$, where $in = s$ is the input, $out^+$ is the correct output, which is a sentence different from $s$ but having the same chunk pattern. $Out^-$ are $N_{negs}$ incorrect outputs, randomly chosen from the sentences that have a chunk pattern different from $s$. The algorithm for building the data and a sample line and generated sentences are shown in appendix A.1.

From the generated instances, we sample uniformly, based on the pattern of the input sentence, approximately 4000 instances, randomly split 80:20 into train:test. The train part is further split 80:20 into train:dev, resulting in a 2576:630:798 split for train:dev:test. We use a French and an English seed file and generate French and English variations of the dataset, with the same statistics.

### 4.2 Blackbird Language Matrices

Blackbird Language Matrices (BLMs) (Merlo, 2023) —language versions of the visual Raven Progressive Matrices (RPMs)— are multiple-choice problems, where the input is a sequence of sentences built using specific generating rules, and the answer set consists of a correct answer that continues the input sequence, and several incorrect contrastive options, built by violating the underlying generating rules of the sentences. In a BLM matrix, all sentences share a targeted linguistic phenomenon, but differ in other aspects relevant for the phenomenon in question. Thus, BLMs, like their visual counterpart RPMs, require identifying the entities (the chunks), their relevant attributes (their morphological or semantic properties) and their connecting operators, to find the correct answer.

To test the detection of different types of information in different languages, we use two BLM datasets, which encode two different linguistic phenomena, each in a different language: (i) BLM-AgrF – subject verb agreement in French (An et al., 2023), and (ii) BLM-s/lE – verb alternations in English (Samo et al., 2023). The structure of these datasets – in terms of the sentence chunks and sequence structure, as well as the answer sets and the erroneous answers and their error types – is shown in Figure 1. Examples are in appendices A.1, A.2.

BLM datasets also have a lexical variation dimension, to explore the impact of lexical variation on detecting relevant structures: type I – minimal lexical variation for sentences within an instance, type II – one word difference across the sentences within an instance, type III – maximal lexical variation within an instance.

The BLM-s/lE dataset is used as is. We built a variation of the BLM-AgrF (An et al., 2023) that separates sequence-based errors (WNA, WN1 and WN2 in Figure 1 – they have correct agreement, but do not respect the pattern of the sequence) from other types of errors, to be able to contrast linguistic errors from errors in identifying sentence parts and

| | Subj.-verb agr | Verb alternations | |
|---|---|---|---|
| | | ALT-ATL | ATL-ALT |
| Type I | 2000:252 | 2000:375 | 2000:375 |
| Type II | 2000:4866 | 2000:1500 | 2000:1500 |
| Type III | 2000:4869 | 2000:1500 | 2000:1500 |

Table 1: Train:Test statistics for the two BLM problems.

understand better how the BLM tasks are solved. The errors in both BLM tasks allow us to study in more detail the performance and understand where the weaknesses are when solving the task.

**Datasets statistics** Table 1 shows the datasets statistics for the BLM problems. After splitting each subset 90:10 into train:test subsets, we randomly sample 2000 instances as train data. 20% of the train data is used for development.

# 5 Experiments

We build upon (Nastase and Merlo, 2023), and use as sentence representations the embedding of the $[CLS]$ special token from a pretrained Electra model (Clark et al., 2020)[2] reshaped as a two-dimensional array. We chose Electra because it has a stronger sentence-level supervision signal as well as strong results on multiple NLU tasks (see Section 2). In Section 5.1.3, we show how it compares to other pretrained models.

The BLM tasks have been benchmarked using FFNN and CNN systems which directly predict the correct answer based on the input sequence (An et al., 2023; Samo et al., 2023). Results improve on both tasks when using a variational encoder-decoder that compresses the input sequence into a very small vector on the latent layer (Nastase and Merlo, 2023). This previous work, and similarity of the BLM tasks with the visual Raven Progressive Matrices task, have led us to a two-step investigation process: (i) using sentences and a VAE-based system, we test whether we can compress sentences into a smaller representation on the latent layer that captures information about the chunk structure of the sentence (Section 5.1 below); (ii) to see if the system can detect and extract the kind of information relevant to a specific task, we combine the compression of the sentence representation with the BLM problems, where a crucial part of the solution lies in identifying the structures of sentences and their sequence in the input (Section 5.2 below). This two-step approach to solving a BLM problem

fits with the way humans solve the visual RPM problems from which the BLMs are inspired: (i) identify the relevant objects and their attributes; (ii) decompose the main problem into subproblems, based on object and attribute identification, in a way that allows detecting the global pattern or underlying rules (Carpenter et al., 1990).

## 5.1 Parts in sentences

We test whether sentence embeddings contain information about the chunk structure of the corresponding sentences by compressing them into a lower dimensional representation in a VAE-like system.

### 5.1.1 Experimental set-up

The architecture of the sentence-level VAE is similar to a previously proposed system (Nastase and Merlo, 2023): the encoder consists of a CNN layer with a 15x15 kernel, which is applied to a 32x24-shaped sentence embedding, followed by a linear layer that compresses the output of the CNN into a latent layer of size 5. The decoder mirrors the encoder, and unpacks a sampled latent vector into a 32x24 sentence representation.

An instance consists of a triple $(in, out^+, Out^-)$, where $in$ is an input sentence with embedding $e_{in}$ and chunk structure $p$, $out^+$ is a sentence with embedding $e_{out+}$ with same chunk structure $p$, and $Out^- = \{s_k | k = 1, N_{negs}\}$ is a set of $N_{negs} = 7$ sentences with embeddings $e_{s_k}$, each with chunk pattern different from $p$ (and different from each other). The input $e_{in}$ is encoded into a latent representation $z_i$, from which we sample a vector $\tilde{z}_i$, which is decoded into the output $\hat{e}_{in}$. We enforce that the latent encodes the structure of the input sentence by using a max-margin loss function, to push for a higher cosine similarity score with the sentence that has the same chunk pattern as the input ($e_{out+}$) than the ones that do not ($E^- = \{e_{s_k} | e_{s_k} = embedding(s_k), s_k \in Out^-\}$).

$$loss_{sent}(e_{in}) = maxM(\hat{e}_{in}, e_{out+}, E^-) + \\ + KL(z_i || \mathcal{N}(0, 1))$$

$$maxM(\hat{e}_{in}, e_{out+}, E^-) = \\ max(0, 1 - cos(\hat{e}_{in}, e_{out+}) + \\ + \frac{\sum_{e_{s_k} \in E^-} cos(\hat{e}_{in}, e_{s_k})}{N_{negs}})$$

At prediction time, the sentence from the $\{out^+\} \cup Out^-$ options that has the highest score relative to the decoded answer is taken as correct.

---

[2] google/electra-base-discriminator

### 5.1.2 Analysis

To assess whether the correct patterns of chunks are detected, we analyze the results for the experiments described in the previous section in two ways: (i) analyze the output of the system, in terms of average F1 score over three runs and confusion matrices; (ii) analyze the latent layer, to determine whether chunk patterns are encoded in the latent vectors (for instance, latent vectors cluster according to the pattern of their corresponding sentences).

In a binary evaluation (has the system built a sentence representation that is closest to the one that has the same chunk pattern as the input?), the system achieves an average positive class F1 score (and standard deviation) over three runs of 0.9992 (0.01) for French, and 0.997 (0.0035) for English.

The pattern-level evaluation for the French data, presented as a confusion matrix based on the pattern information for $out^+, Out^-$ at the top of Figure 2, shows that all patterns are detected with high accuracy (the results for English are in Appendix A.4.2). To understand how chunk information is encoded on the latent layer, we perform latent traversals: for each instance in the test data, after encoding it, we modify the value of each unit in the latent layer with ten values in its min-max range, based on the training data, and decode the answer.

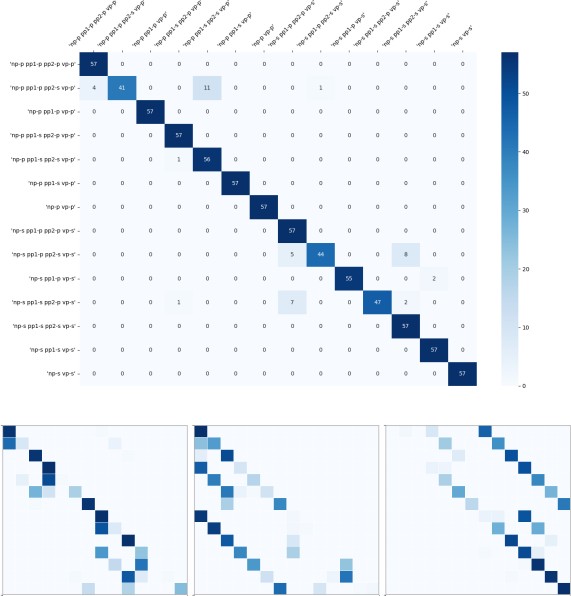

Figure 2: Latent layer encoding of pattern information: **top** confusion matrix for pattern-level evaluation; **bottom** sample of effects of latent traversal in terms of pattern-level evaluation.

The confusion matrices presented as heatmaps in the bottom part of Figure 2 (a larger version in Figure 10 in Appendix A.4) show that specific changes to the latent vectors decrease the differentiation among patterns, as expected if chunk pattern information were encoded in the latent vectors. Changes to latent unit 1 cause patterns that differ in the grammatical number of $pp2$ not to be distinguishable (left matrix). Changes to latent units 2 and 3 lead to the matrices in the middle and right of the figure, where patterns that have different subject-verb grammatical number are indistinguishable.

To confirm that chunk information is present in the latent layer, we plot the projection of the latent vectors in two dimensions (Figure 3). The plot shows a very crisp clustering of latents that correspond to input sentences with the same chunk pattern, despite the fact that some patterns differ by only one attribute (the grammatical number) of one chunk.

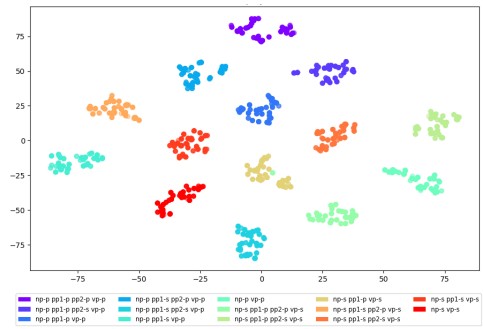

Figure 3: Chunk identification: tSNE projections of the latent vectors for the French dataset.

### 5.1.3 Electra vs. BERT and RoBERTa, and the price of fine-tuning

There are differences in the architectures, training objectives and training data for transformer-based models, which lead to differences in how they encode information. Fine-tuning further changes the landscape of the embeddings, and prioritizes different characteristics of the input sentence, often semantics. We can quantify some of these differences using the setup described above.

Experiments on the task of reconstructing a sentence with the same chunk structure on BERT[3] (Devlin et al., 2019) and RoBERTa[4] (Liu et al., 2019) lead to average F1 score over 3 runs of 0.91 (std=0.0346) for BERT and 0.8926 (std=0.0166)

---

[3] https://huggingface.co/google-bert/bert-base-multilingual-cased
[4] https://huggingface.co/FacebookAI/xlm-roberta-base

for RoBERTa, confirming that Electra's architecture leads to sentence embeddings that encode more explicitly structure-related information.

Two sentence transformer models LaBSE and MPNet[5] obtained an average F1 of 0.43 (std=0.0336) and 0.669 (std=0.0407) respectively. We chose LaBSE and MPNet because they are tuned differently – LaBSE is trained with bilingual sentence pairs with high results on a cross-language sentence retrieval task, MPNet is optimized for sentence similarity – and their representations have the same dimensionality (768) as the transformer models we used. The low results on detecting chunk structure in sentence embeddings after this tuning indicates that in the quest of optimizing the representation of the meaning of a sentence, structural information is lost.

## 5.2 Parts in sentences for BLM tasks

We test whether including the sentence compression step in a system to solve the BLM tasks leads to latent representations that contain information about chunk properties relevant to the tasks.

### 5.2.1 Experimental setup

The BLM problems encode a linguistic phenomenon in a sequence of sentences that have regular and relevant structure, which serves to emphasize and reinforce the encoded phenomenon. (Carpenter et al., 1990). We model the process of solving a BLM in a manner similar to how humans solve RPM visual tasks, by using the two-level intertwined architecture illustrated in Figure 4: one level for detecting sentence structure, one for detecting the correct answer based on the sentence structure and their sequence.

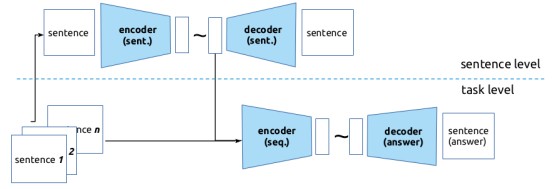

Figure 4: A two-level VAE-based system: the sentence level learns to compress a sentence into a representation useful to solve the BLM problem on the task level.

An instance for a BLM problem consists of an ordered sequence $S$ of sentences, $S = \{s_i | i = 1, 7\}$

[5]https://huggingface.co/sentence-transformers/LaBSE, https://huggingface.co/sentence-transformers/all-mpnet-base-v2

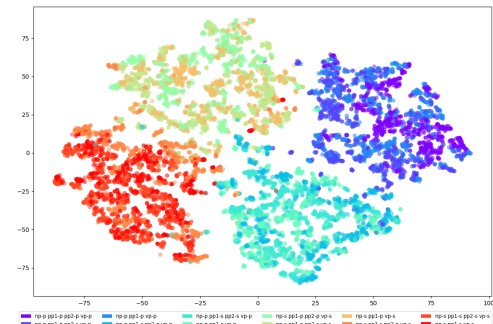

**a.)** TSNE projection of latent representations from the latent layer of the sentence level for the sentences in BLM contexts in the training data, coloured by the chunk pattern.

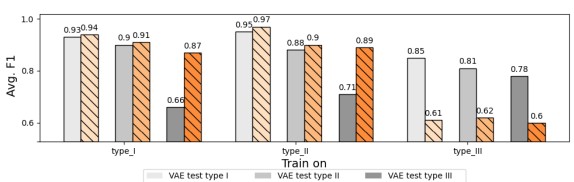

**b.)** Average F1 score over 3 runs, grouped by training data on the x-axis, tested on type I, II, III in different shades.

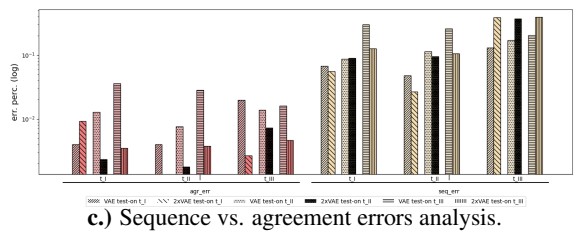

**c.)** Sequence vs. agreement errors analysis.

Figure 5: VAE vs 2-level VAE (2xVAE) on the agreement BLM problem

as input, and an answer set $A$ with one correct answer $a_c$, and several incorrect answers $a_{err_j}$. The sentences in $S$ are passed as input to the sentence-level VAE, which is the system described in Section 5.1. The latent representations from this VAE are used as the representations of the sentences in $S$. These representations are passed as input to the BLM-level VAE, in the same order as $S$. From the compressed layer of the BLM-level VAE, the decoder reconstructs a sentence embedding ($e_S$), which is compared to the embeddings of the answers.

An instance for the sentence-level VAE consists of a triple $(s_i, out_i^+, Out_i^-)$. For our two-level system, we must construct this triple on the fly from the input BLM instance: $s_i \in S$ with embedding $e_{s_i}$, $out_i^+ = s_i$, and $Out_i^- = \{s_k | s_k \in S, s_k \neq s_i\}$ with embeddings $E_i^- = \{e_{s_k} | k = 1, N_{negs}\}$. The loss combines the loss signal from the two levels:

$$loss(S) = \sum_{s_i \in S} loss_{sent}(e_{s_i}) + loss_{task}(e_S)$$

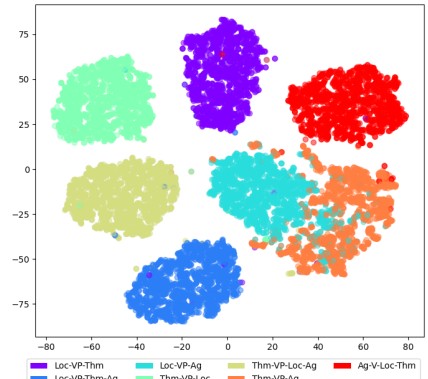

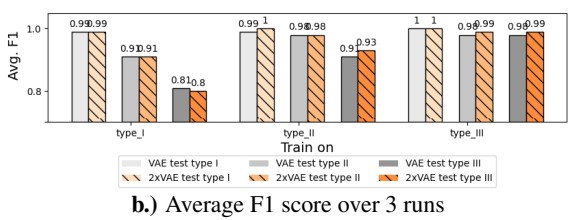

**a.)** TSNE projection of latent representations from the latent layer of the sentence level for the sentences in BLM contexts in the training data, coloured by the pattern of semantic roles.

**b.)** Average F1 score over 3 runs

Figure 6: VAE vs 2-level VAE (2xVAE) on the verb alternation BLM problem, Group 1 (Agent-Location-Theme -> Agent-Theme-Location)

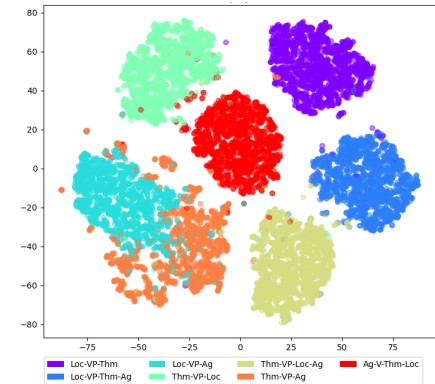

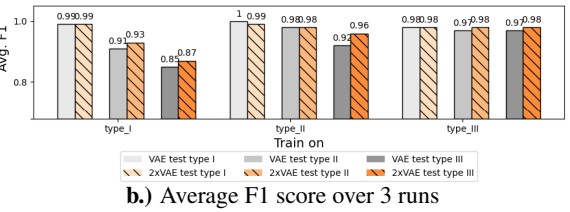

**a.)** TSNE projection of latent representations from the latent layer of the sentence level for the sentences in BLM contexts in the training data, coloured by the pattern of semantic roles.

**b.)** Average F1 score over 3 runs

Figure 7: VAE vs 2-level VAE (2xVAE) on the verb alternation BLM problem, Group 2 (Agent-Theme-Location -> Agent-Location-Theme)

The loss at the sentence level is computed as described in Section 5.1:

$$loss_{sent}(e_{s_i}) = maxM(e_{s_i}, e_{out_i^+}, E_i^-)$$
$$+ KL(z_i | \mathcal{N}(0,1))$$

The loss at the task level is computed in a similar manner, but relative to the answer set $\mathcal{A}$ with the corresponding embeddings set $E_A$, and the correct answer $a_c$, of the task:

$$loss_{task}(e_S) = maxM(e_S, e_{a_c}, E_A \setminus e_{a_c})$$
$$+ KL_{seq}(z_S | \mathcal{N}(0,1)).$$

### 5.2.2 Analysis

We run experiments on the BLMs for agreement (Figure 5) and for verb alternation (Figures 6, 7), to test a range of syntactic and semantic chunk properties that should be identified. While the information necessary to solve the agreement task is more structural, solving the verb alternation task requires both structural information concerning chunks and semantic information, with syntactically similar chunks playing different roles in a sentence (see Figure 1). The results show that the two-level sys-

tem leads to better results compared to the one-level process for these structure-based linguistic problems, thereby providing additional support to our hypothesis that chunks and their attributes are detectable in sentence embeddings.

The results in terms of average F1 scores for the agreement task, and the latent representation and analysis of the errors made by the system are shown in Figure 5, and provide several insights. Detailed results are in the appendix.

First, the latent representation analysis (Figure 5.a) shows that while the sentence representations on the latent layer are not as crisply separated by their chunk pattern as for the experiment in Section 5.1, there is a clear separation in terms of the grammatical number of the subject and the verb. This is not surprising as the focus of the task is subject-verb agreement. However, as shown by the results in term of F1 (Figure 5.b) and the analysis of the errors made by the system on the task (Figure 5.c, and more detailed in Figure 12 in Appendix A.5.3), there is enough information in these compressed latent representations to capture the structural regularities imposed by the patterns of chunks in the input sequence.

Second, the results in terms of F1 (Figure 5.b) show that the two-level process generalizes better

from simpler data – learning on type I and type II leads to better results on all test data, with the highest improvement when tested on type III data, which has the highest lexical variation. Furthermore, the two-level models learned when training on the lexically simpler data perform better when tested on the type III data than the models learned on type III data itself. This result not only indicates that structure information is more easily detectable when lexical variation is less of a factor, but more importantly, that chunk information is separable from other types of information in the sentence embedding, as the patterns detecting it can be applied successfully for data with additional (lexical) variation.[6]

The analysis of the errors made by the system (Figure 5.c) shows that the two-level system has a lower rate of sequence errors (WNA, WN1, WN2 – see Figure 1), which from the point of view of the targeted phenomenon are correct (see Section 4.2). The fact that without the sentence compression step (using the one-level model) the system makes more sequence-based errors, indicates that modeling structural information separately is not only possible, but also beneficial for some tasks.

The results on the verb alternation BLMs are shown in Figures 6 and 7. In this problem, structurally similar chunks - NPs, PPs – play different semantic roles in the verb alternation data, as shown in Figure 1. The TSNE projection of the latent representations on the sentence level (Figures 6.a, 7.a) and the F1 results on the task (Figures 6.b, 7.b) show that the system is able to detect such syntactic-semantic information in the sentence embeddings. The closest latent representations are two that have the same syntactic pattern: *NP VerbPass PP*, but differ semantically: *NP-Theme VerbPass PP-Agent* vs. *NP-Loc VerbPass PP-Agent*, yet they are still distinguished. Detailed error results are included in Figure 13 in Appendix A.5.3.

### 5.3 Discussion

We performed two types of experiments: (i) use individual sentences, and an indirect supervision signal about the sentence structure, (ii) incorporate a sentence representation compression step in a task-specific setting. We have used two tasks, one which relies on more structural information (subject-verb agreement), and one that also relies on semantic information about the chunks (verb alternation).

We investigated each setup by the results on the task – average F1 scores, and analysis of the type of errors made by the system (as described in Figure 1) – and by the compressed sentence representations on the latent layer of an encoder-decoder architecture.

By this dual analysis, one can conclude not only whether a task is solved correctly, but also whether it is solved using structural, morphological and semantic information from the sentence. We found that information about (varying numbers of) chunks – noun, verb and prepositional phrases – and their task-relevant attributes, morphological or semantic, can be detected in sentence embeddings from a pretrained transformer model.

The use of probes has been questioned, as the probe itself may assemble the requested information without detecting or modeling the phenomenon of interest (Hewitt and Liang, 2019; Belinkov, 2022; Lenci, 2023). To partially address this problem, we have used only indirect supervision – within the system, there is no direct information about what characteristics of the answer (on the sentence or the task level) are relevant. Despite the lack of direct supervision, the system is able to compress the structural information necessary to solve the task onto the latent layer of the sentence encoder. In future work, we will investigate whether this information is "hard-coded" – encoded consistently across languages and tasks – in the embeddings, or it relies on shallower features.

## 6  Conclusions

Sentence embeddings obtained from transformer models are compact representations, compressing much knowledge —morphological, grammatical, semantic—, expressed in text fragments of various length, into a vector of real numbers of fixed length. We can separate this representation into different layers using a convolutional neural network and distinguish specific information among these layers. In particular, we have shown that we can detect information about chunks – noun/verb/prepositional phrases – and their task-relevant attributes, without providing direct supervision to the system about the targeted structures. This brings us one step closer to understanding and unpacking transformer-based sentence embeddings.

---

[6]Explanation in Appendix A.5.1

# 7 Limitations

We have performed experiments on datasets containing sentences with specific structure and properties to be able to determine whether the type of information we targeted can be detected in sentence embeddings. We have used this data to avoid directly training a classifier, which may learn the task of distinguishing sentences with different chunk patterns without actually using such information from the sentence embeddings. Despite our analyses, there is no guarantee that the information about chunks and their properties is not assembled on the fly from more fine-grained information in the sentence embedding. In future work we plan to investigate whether this is the case, or whether what is encoded is something more abstract, akin to a rule.

**Acknowledgments**    We gratefully acknowledge the support of this work by the Swiss National Science Foundation, through grant SNF Advanced grant TMAG-1_209426 to PM.

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

# A Appendix

## A.1 Sentence data

To build the sentence data, we use a seed file that was used to generate the subject-verb agreement data. A seed, consisting of noun, prepositional and verb phrases with different grammatical numbers, can be combined to build sentences consisting of different sequences of such chunks. Table 2 includes a partial line from the seed file, from which individual sentences and a BLM instance can be constructed. We use French and English versions of the seed file to build the corresponding datasets.

| Subj_sg | Subj_pl | P1_sg | P1_pl | P2_sg | P2_pl | V_sg | V_pl |
|---|---|---|---|---|---|---|---|
| The computer | The computers | with the program | with the programs | of the experiment | of the experiments | is broken | are broken |

| **Sent. with different chunks** | | **a BLM instance** |
|---|---|---|
| | | Context: |
| | | The computer with the program is broken. |
| The computer is broken. | np-s vp-s | The computers with the program are broken. |
| | | The computer with the programs is broken. |
| The computers are broken. | np-p vp-p | The computers with the programs are broken. |
| | | The computer with the program of the experiment is broken. |
| The computer with the program is broken. | np-s pp1-s vp-s | The computers with the program of the experiment are broken. |
| | | The computer with the programs of the experiment is broken. |
| ... | ... | Answer set: |
| | | *The computers with the programs of the experiment are broken.* |
| The computers with the programs of the experiments are broken. | np-p pp1-p pp2-p vp-p | The computers with the programs of the experiments are broken. |
| | | The computers with the program of the experiment are broken. |
| | | The computers with the program of the experiment is broken. |
| | | ... |

Table 2: A line from the seed file on top, and a set of individual sentences built from it, as well as one BLM instance.

The algorithm to produce a dataset from the generated sentences is detailed in Figure 8 below.

```
Data = []; N_negs
for patterns p do
    for s_i ∈ S_p do
        in = s_i
        for s_j ∈ S_p do
            out^+ = s_j
            out^- = {s_k, k ∈ range(N_negs), s_k ∈ S_¬p}
            Data = Data ∪ [(in, out^+, out^-)]
        end for
    end for
end for
```

Figure 8: Data generation algorithm

## A.2 Example of data for the verb alternation BLM

TYPE I

| EXAMPLE OF CONTEXT |
| --- |
| The buyer can load the tools in bags. |
| The tools were loaded by the buyer |
| The tools were loaded in bags by the buyer |
| The tools were loaded in bags |
| Bags were loaded by the buyer |
| Bags were loaded with the tools by the buyer |
| Bags were loaded with the tools |
| ??? |

| EXAMPLE OF ANSWERS |
| --- |
| **The buyer can load bags with the tools** |
| The buyer was loaded bags with the tools |
| The buyer can load bags the tools |
| The buyer can load in bags with the tools |
| The buyer can load bags on sale |
| The buyer can load bags under the tools |
| Bags can load the buyer with the tools |
| The tools can load the buyer in bags |
| Bags can load the tools in the buyer |

Figure 9: Example of Type I context sentences and answer set.

## A.3 Experimental details

All systems used a learning rate of 0.001 and Adam optimizer, and batch size 100. The system was trained for 300 epochs for all experiments.

The experiments were run on an HP PAIR Workstation Z4 G4 MT, with an Intel Xeon W-2255 processor, 64G RAM, and a MSI GeForce RTX 3090 VENTUS 3X OC 24G GDDR6X GPU.

The **sentence-level encoder decoder** has 106 603 parameters. It consists of an encoder with a CNN layer followed by a FFNN layer. The CNN input has shape 32x24. We use a kernel size 15x15 with stride 1x1, and 40 channels. The linearized CNN output has 240 units, which the FFNN compresses into the latent layer of size 5+5 (mean+std). The decoder is a mirror of the encoder, which expands a sampled latent of size 5 into a 32x24 representation.

The **two-level system** consists of the sentence level encoder-decoder described above, and a task-specific layer. The input to the task layer is a 7x5 input (sequence of 7 sentences, whose representation we obtain from the latent of the sentence level), which is compressed using a CNN with kernel 4x4 and stride 1x1 and 32 channels into ... units, which are compressed using a FFNN layer into a latent layer of size 5+5 (mean+std). The decoder consists of a FFNN which expands the sampled latent of size 5 into 7200 units, which are then processed through a CNN with kernel size 15x15 and stride 1x1, and produces a sentence embedding of size 32x24. The two level system has 178 126 parameters.

## A.4 Sentence-level analysis

### A.4.1 Sample confusion matrices for altered latent values

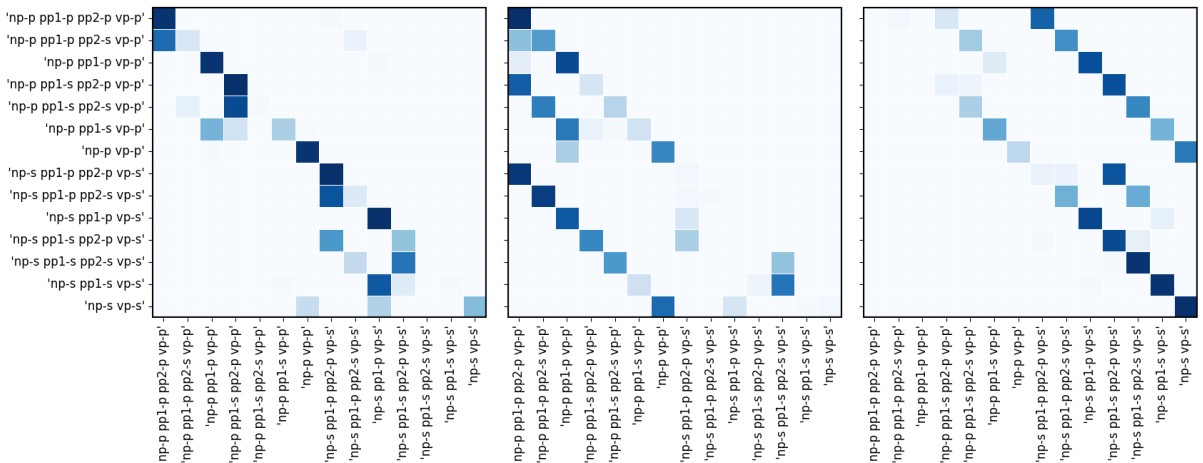

Figure 10: Confusion matrices for altered values on units 1 (left matrix), unit 2 (middle matrix) and unit 3 (right matrix)

Each matrix shows a particular way of conflating different patterns:

- changes to values in unit 1 of the latent lead to patterns that differ in the grammatical number of $pp2$ to become indistinguishable

- changes to values in units 2 and 3 of the latent lead to the conflation of patterns that have different subject-verb numbers.

## A.4.2    Sentence-level analysis for English data

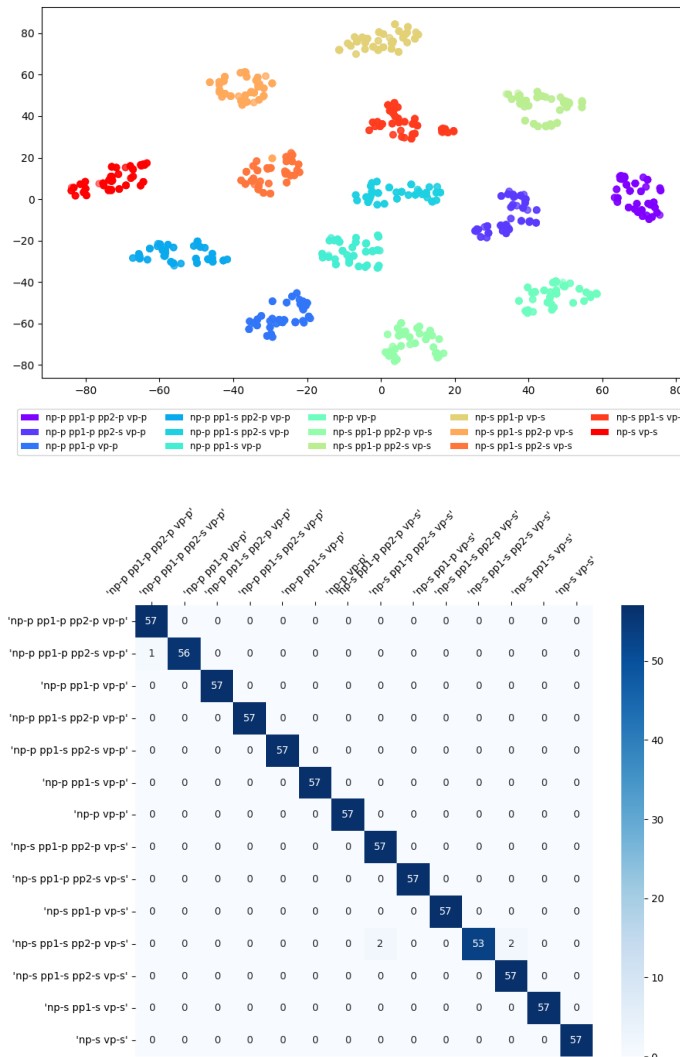

Figure 11: Chunk identification results: tSNE projections of the latent vectors for the English dataset, and confusion matrix of the system output.

## A.5 The BLM tasks

### A.5.1 Discussion of errors on the sentence level, when solving the BLM task

It might appear surprising that the two-level approach leads to lower performance on type III data, particularly when lexical variation had not been an issue for the sentence representation analysis (see Section 5.1).

The difference comes from the way the instances were formed, on the fly, for the two-level process. The only input to the system is the input of the task. This input, consisting of a sequence of 7 sentences, is used to generate an instance – i.e. a $(in, out^+, Out^-)$ triple – for the sentence level process for each of these sentences. Because each sentence has a different pattern, and the input and correct output of the sentence level VAE must have the same pattern, the only possible $out_+$ is the input sentence $in$ itself. $Out^-$ will consist of all the other sentences in the task input sequence.

We hypothesize that the fact that the input and output are identical weakens the (indirect) supervision signal. In the stand-alone sentence analysis experiment, the lexical variation between the input and correct answer for the sentence level forces the system to find deeper shared information between the two, and this is not the case when solving the BLM tasks with the two-level system. For type I and type II data, because a task instance (and thus the input sequence) has very little lexical variation, the incorrect answers for the sentence level are very close lexically to the correct answer, and thus the system is guided to encode on the latent layer other distinctions between the correct and incorrect answers, which are mainly the chunk patterns. For type III data, with its maximal lexical variation, there is no pressure on the system to find something other than shallower differences between the answer candidates.

We plan to test this hypothesis in future work using a pre-trained sentence-level VAE.

## A.5.2 Detailed task results

| TRAIN ON | TEST ON | VAE | 2 LEVEL VAE |
|---|---|---|---|
| **BLM agreement** | | | |
| type_I | type_I | 0.929 (0) | **0.935** (0.0049) |
| type_I | type_II | 0.899 (0) | **0.908** (0.0059) |
| type_I | type_III | 0.662 (0) | **0.871** (0.0092) |
| type_II | type_I | 0.948 ($<$e-10) | **0.974** (0.0049) |
| type_II | type_II | 0.879 ($<$e-10) | **0.904** (0.0021) |
| type_II | type_III | 0.713 (0) | **0.891** (0.0015) |
| type_III | type_I | **0.851** (0.037) | 0.611 (0.1268) |
| type_III | type_II | **0.815** (0.0308) | 0.620 (0.1304) |
| type_III | type_III | **0.779** (0.0285) | 0.602 (0.1195) |
| | | | |
| **BLM verb alternation group 1** | | | |
| type_I | type_I | 0.989 (0) | **0.995** ($<$e-10) |
| type_I | type_II | 0.907 (0) | **0.912** (0.0141) |
| type_I | type_III | **0.809** (0) | 0.804 (0.0167) |
| type_II | type_I | 0.989 (0) | **0.996** (0.0013) |
| type_II | type_II | 0.979 ($<$e-10) | **0.984** (0.0016) |
| type_II | type_III | 0.915 (0) | **0.928** (0.0178) |
| type_III | type_I | 0.997 (0) | **0.999** (0.0013) |
| type_III | type_II | 0.977 (0) | **0.986** (0.0027) |
| type_III | type_III | 0.98 (0) | **0.989** (0.0003) |
| | | | |
| **BLM verb alternation group 2** | | | |
| type_I | type_I | **0.992** (0) | 0.987 (0.0033) |
| type_I | type_II | 0.911 (0) | **0.931** (0.0065) |
| type_I | type_III | 0.847 (0) | **0.869** (0.0102) |
| type_II | type_I | **0.997** (0) | 0.993 (0.0025) |
| type_II | type_II | **0.978** ($<$e-10) | **0.978** (0.0017) |
| type_II | type_III | 0.923 (0) | **0.956** (0.0023) |
| type_III | type_I | 0.979 ($<$e-10) | **0.981** (0.0022) |
| type_III | type_II | 0.972 (0) | **0.975** (0.0005) |
| type_III | type_III | 0.967 (0) | **0.977** (0.0022) |

Table 3: Analysis of systems: average F1 (std) scores (over 3 runs) for the VAE and 2xVAE systems. The highest value for each train/test combination highlighted in bold.

### A.5.3 Detailed error results

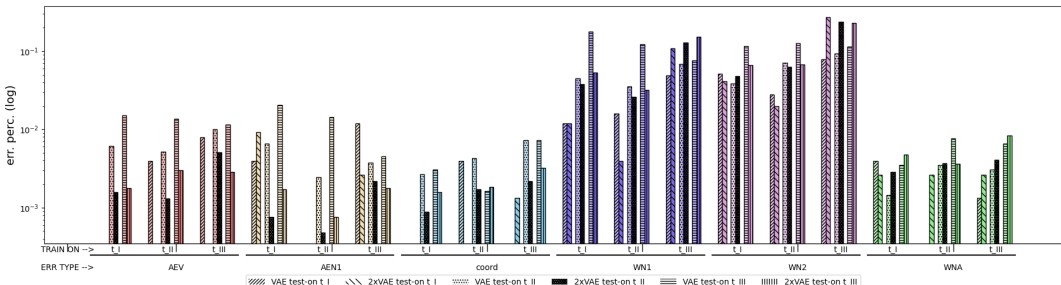

Figure 12: Analysis of errors for the agreement task: y-axis is the log of error percentages, the x-axis indicates the data type the system was trained on. The bars show the errors for testing using the two system variations (one-level and two-level), and the test data type. We note a decrease in all types of errors for the 2-level system compared to the one level version, and particularly for the sequence-based errors (WNA, WN1, WN2) which are overall the most frequent. The reason for the higher number of sequence errors for the system trained on type III data is discussed in appendix A.5.3.

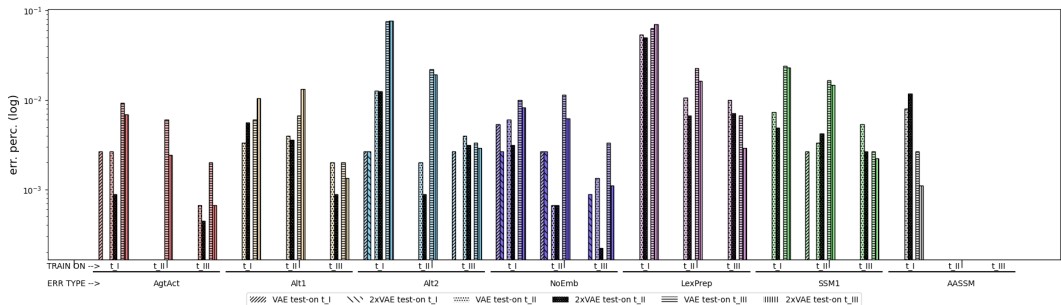

Analysis of errors for the verb alternation group1 task: y-axis is the log of error percentages, the x-axis indicates the data type the system was trained on. The bars show the errors for testing using the two system variations (one-level and two-level), and the test data type. As for the agreement task, we note a decrease in all types of errors when using the 2-level system compared to the one level version, with a few exceptions – Alt1 (a syntactic error) when training on data type I and testing on types II and III.

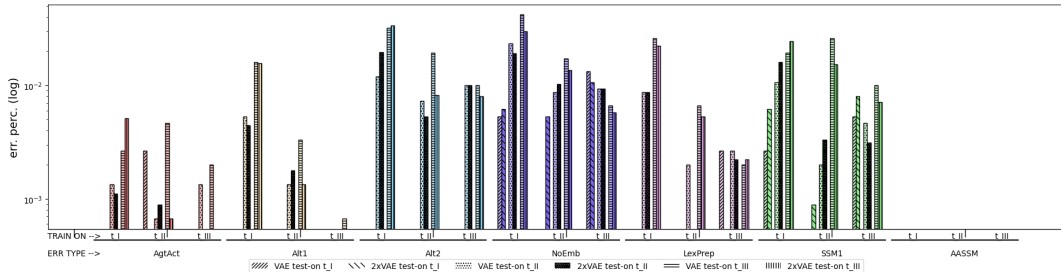

Analysis of errors for the verb alternation group2 task: y-axis is the log of error percentages, the x-axis indicates the data type the system was trained on. The bars show the errors for testing using the two system variations (one-level and two-level), and the test data type. As for the agreement task, we note a decrease in all types of errors when using the 2-level system compared to the one level version, with a few exceptions – SSM1 (a syntax-semantic mapping error), and a few combinations of training/test data types for the syntactic errors Alt1,Alt2.

Figure 13: Error analysis for the verb alternation BLM task.

