# OpenReview forum: "Are there identifiable structural parts in the sentence embedding whole?"
_EMNLP/2024/Workshop/BlackBoxNLP — BlackboxNLP 2024_

### Official Review · Reviewer_utLK · 2024-09-08

**Overall Assessment:** 3
**Confidence:** 3

**Best Paper:**

1

**Best Paper Justification:**

N/A

**Comments Questions Suggestions And Typos:**

- “Transformer architectures compress the informa021 tion in a sentence – morphological, grammati022 cal, semantic, pragmatic – into a fixed-length one023 dimensional array of real numbers” -- Should be adjusted. This is only one thing you can do with Transformers. They output a sequence of contextualized embeddings. One thing you can do with these embeddings is compress them into a single vector, for example if you use the <CLS> token as input to a classifier. But this is not what the GPTs do, for example.

- Suggestion: Link the VAE approach to SAEs, which are recently extremely hot as an interpretability technique.

"We hypothesize that different types of information from an input are melded together, and no longer overtly accessible in the sentence embeddings" - should be qualified, what is meant is linear accessibility.

**Paper Summary:**

The paper asks whether structural parts of sentences---that is, chunks---are present in sentence embeddings. The paper uses Blackbird Language Matrices, a clever form of Raven's Progressive Matrices where the features that are patterned are grammatical features rather than visual features.

**Summary Of Strengths:**

- The paper asks an interesting question
- The BLM approach is clever and gets at linguistic representation and reasoning simultaneously
- Results are presented carefully

**Summary Of Weaknesses:**

- The approach is somewhat convoluted. If I was interested in the research question posed by the title, my first instinct would be to use methods like PCA and ICA to try to find components in the sentence embedding and see if they correspond to sentence parts. A number of other simpler ideas come to mind. Even regarding the BLM task, the method of training a two-level VAE is not as simple as the first thing I would think of, which would be just to provide a BLM pattern as a prompt for an autoregressive model.
- As a result of the unorthodox methods, the results are hard for me to interpret.
- The work focuses on BERT-style models which are not so relevant to modern LLMs. The same methods could easily be applied to a GPT, but aren't.
- In principle it doesn't seem that reshaping the sentence embedding should not make information any more or less accessible. This suggests that there is something amiss in the approach.

---

### Official Review · Reviewer_Qe7e · 2024-09-08

**Overall Assessment:** 2
**Confidence:** 4

**Best Paper:**

1

**Best Paper Justification:**

-

**Comments Questions Suggestions And Typos:**

Consider making the paper more clear to follow by significantly cutting down on content and simplifying it. Now there are too many graphs and a lot of text that do not really add value to the main argument. Present the results more clearly and consider the role of each graph and visualization.

**Paper Summary:**

The paper examines transformer-based sentence embedding models and whether the models encode structural information such as parts of speech or phrase structures.

**Summary Of Strengths:**

* The research topic, question and the experiments are very interesting and worthy of more research
* The paper contains a lot of useful references to relevant research.
* The paper indicates that syntactic information is encoded in transformer models and can be detected from them.

**Summary Of Weaknesses:**

* Although the research question and the premise for the paper are very interesting and worthy of more research, the paper is quite complex and difficult to follow.
* The paper uses a lot of abbreviations that are well known for practitioners and researchers working with the specific types of models and tasks. However, in such a paper they should be explained. For example: FFNN, CNN, RPM, VAE etc.
* The graphs and plots are very small with text that is impossible to read. IT is difficult to understand the purpose and the relevance of each graph as the captions are fairly uninformative. E.g. *"Figure 5: VAE vs 2-level VAE (2xVAE) on the agreement BLM problem"*
* The experimental results are difficult to interpret from the paper. The results are not clearly presented.
* The paper refers to error analysis (e.g. line 529), however there is no real error analysis included in the paper.

---

### Official Review · Reviewer_pXAv · 2024-09-09

**Overall Assessment:** 4
**Confidence:** 3

**Best Paper:**

1

**Best Paper Justification:**

n/a

**Comments Questions Suggestions And Typos:**

It would be neat to see the similar results for sentence-transformers you have for the first chunk task for BLM! You should expect them to do much much worse than ELECTRA or BERT right?

- More of a writing comment than anything regarding context, the definition of chunk should probably be moved earlier in the text to help with readability.
- extra parenthesis line 27 page 1
- missing number in line 948 in the appendix
- The references to section numbers in lines 194 to 203 page 3 are a bit confusing
- space after comma line 344 page 5

**Paper Summary:**

This paper uses a VAE model to analyse the syntactic structure that can be gleamed from sentence embeddings. Specifically, the authors look at the classification token, "[CLS]" of BERT-style encoder models (as well as a brief look at semantically trained models, e.g. SentenceTransformers). Specifically there are two main tasks:

i) A simple VAE trained on sentences constructed from a specific syntactic recipe NP (PP (PP)) VP. () here meaning optional. (It's not entirely clear what characterized the NPs or VPs (presumably they did not include PPs themselves)).  The model takes a 768 dimensional sentence embedding and re-organises it into a 32x24 matrix and then passes it through a 1 layer CNN into a latent dimension of only 5 after which it is passed to the encoder which upscales it to 32x24 again. The loss was trained to maximise the similarity of the output vector to the input vector while getting as far as possible from other sampled vectors (which crucially didn't have the same syntactic parse).

The trained VAE is close to the target vector in the vast majority of cases (rather unsurprisingly) but interestingly the latent dimension can be perturbed and systematic (e.g. latent units 2-3 encode grammatical number for subject verb agreement). The latent representations also seem highly separable along syntactic lines, even when differences are subtle (pl vs sg agreement) and the sentences are semantically quite different (fig 3). Most interestingly, sentence transformers which are trained on semantic similarity of sentences rather than BERT style cloze tasks or ELECTRA's pretraining paradigm perform much much worse and separating different syntactic patterns.

ii) The second task uses a similar VAE model to do Blackbird Language Matrices (a syntactic analogy of Raven progressive matrices), here the model must choose the best answer out of a set of possible answers given a context of preceding sentences which match to one sentence's syntactic structure. This model contains two VAEs. The first is akin to (i) and is trained by first taking the latent representations of all sentences in a BLM sequence and then trying to reconstruct the latent representations for each sentence while maximising the difference to the other sentences in the BLM sequence. The second then takes these latent representations for all 7 sentences in the sequence and then tries to reconstruct the answer sentence embedding while keeping it far from the wrong answers in the answer set.

Under this approach, the different syntactic structures are not as easily separable in the latent representation. However, the model still largely accurate at choosing the answer sentence in a selection of BLM tasks and is much better than the single layer VAE (except on "type iii" BLMs where the different sentences have maximal lexical variation rather than largely using the same words).

**Summary Of Strengths:**

- The use of VAE model for probing in a semi-supervised manner is quite interesting.
- The intepretability of some of the latent units is super cool and I think a very good sign that relatively high level syntactic information is encoded in [cls] vectors. The fact that sentence transformers are also much much worse at this task than BERT-based or ELECTRA-based models is very cool as well and I think a good sign for the approach.
 - The fact the separability of different syntactic structures is possible even in much less constrained cases  (e.g. BLM) is also a good sign and shows the patterns are probably easily found in the structure of the sentence embeddings and the fact the some of the tasks are solvable seems to demonstrate that the syntactic information is encoded.
- The fact that the single vae model does much worse when trained on lexically diverse sentence embeddings (type iii BLM) compared to the 2x VAE model is interesting and makes sense as otherwise the syntactic information is not as "salient" for the model.

**Summary Of Weaknesses:**

- I'm not completely sold that the VAE method really resolves the problems with probing, I think a bit more argumentation in that regard could be useful for the paper.
- The writing is a little opaque, I think the paper could benefit from a bit of restructuring (i.e. being more specific about the tasks done in the first page or two and perhaps a bit less literature review in favour of expansion of explanation of the two-level model)
- In particular, this lack of clarity is particularly problematic with regard to the metrics used. The writing seems ambiguous for two important points:

i) For Figure 2 and the F1 scores in section 5.1.3 is the model evaluated on choosing the *input* sentence or a sentence with the same syntactic structure as the input sentence? The writing seems compatible with both possibilities and it's crucial to be clear as to which it is. (Likewise it would be good to explicitly state that the F1 scores in figure 5 are on the BLM tasks described in figure 1)

ii) For the 2x VAE model: I initially read this part very differently, understanding it to be that the sentence-level model was trained like the original VAE model and then the sequence level was trained by contrasting the triple constructed from the example sentence members. I think the writing in this paragraph could be greatly improved by just describing the loss function a bit more clearly: it would be good to explicitly show the triple used for sequence level and sentence level side-by-side.

---

### Decision · Program_Chairs · 2024-09-18

**Decision:**

Accept

**Comment:**

While reviewers found the research question and the experiments interesting, they unanimously had trouble following some parts of the paper. The authors should incorporate the reviewers' comments to improve the writing and presentation of the paper, and expand the informativeness of figure captions for the camera-ready version.